# First-Trimester Screening for Gestational Diabetes Mellitus in Twin Pregnancies

**DOI:** 10.3390/jcm10173814

**Published:** 2021-08-25

**Authors:** Olga Buerger, Tania Elger, Antonia Varthaliti, Argyro Syngelaki, Alan Wright, Kypros H. Nicolaides

**Affiliations:** 1Fetal Medicine Research Institute, King’s College Hospital, London SE5 8BB, UK; olga.buerger@nhs.net (O.B.); tania.elger@nhs.net (T.E.); antonia.varthaliti@nhs.net (A.V.); argyro.syngelaki@nhs.net (A.S.); 2Institute of Health Research, University of Exeter, Exeter EX4 4QG, UK; alan@dw-stats.co.uk

**Keywords:** first trimester screening, pyramid of prenatal care, gestational diabetes mellitus, twin pregnancies

## Abstract

We previously reported a logistic regression model for prediction of GDM from maternal characteristics and medical history in 75,161 singleton pregnancies. In this study of 1376 twin and 13,760 singleton pregnancies recruited at 11–13 weeks’ gestation, we extend the model to include terms for twin pregnancies. We found the respective odds of GDM in dichorionic and monochorionic twin pregnancies to be 1.36 (95% CI: 1.02–1.81) and 2.78 (95% CI: 1.72–4.48) times higher than in singleton pregnancies. In both singleton and twin pregnancies, the risk for GDM increased with maternal age and weight and birth weight z-score of a baby in a previous pregnancy and is higher in women with a previous pregnancy complicated by GDM; in those with a first- or second-degree relative with diabetes mellitus; in women of Black, East Asian, and South Asian racial origin; and in pregnancies conceived through the use of ovulation-induction drugs. In singleton pregnancies, at 10% and 20% false-positive rate, the detection rate was 43% and 58%, respectively. In twin pregnancies, using risk cut-offs corresponding to 10% and 20% false-positive rates in singletons, the respective false-positive rates were 27% and 47%, and the detection rates were 63% and 81%.

## 1. Introduction

Gestational diabetes mellitus (GDM) is associated with a plethora of complications for both mother and baby in short and long term [1,2,3,4,5,6]. The diagnosis of GDM relies on a positive oral glucose tolerance test, which can be performed either universally to all pregnant women [7] or selectively depending on certain risk factors from demographic characteristics and obstetric history [8]. In a previous study of 75,161 singleton pregnancies, we developed a screening model for GDM by combining various maternal characteristics and history through multivariate logistic analysis, and we compared the screening performance of our model with that recommended by National Institute of Health and Care Excellence (NICE) [8,9]. Our study showed that the performance of screening for GDM was superior when using a multivariate logistic model rather than treating each maternal factor as an independent screening test [8,9]. The significant contributors in our prediction model for GDM included maternal age, weight, height, racial origin, family history of diabetes, use of ovulation drugs, birth weight, and history of GDM in a previous pregnancy.

Twin pregnancies, compared to singletons, are at increased risk of pregnancy complications, including miscarriage, stillbirth, preterm birth, fetal growth restriction, and preeclampsia [10,11,12]. There is contradictory evidence concerning the incidence of GDM in twin pregnancies, with some studies reporting that this is higher than in singletons [13,14,15,16,17,18], while others found that the rates are similar [17,18]. 

The objectives of this study were to: first, compare the rate of GDM in dichorionic and monochorionic twin pregnancies with that in singleton pregnancies and second, to modify our previously reported prediction model for GDM in singleton pregnancies [9] to include twin pregnancies.

## 2. Materials and Methods

### 2.1. Study Population and Design

The study population was derived from a prospective screening study on the early prediction of pregnancy complications in women attending for their routine first hospital pregnancy visit at King’s College Hospital, London, between January 2010 and August 2020. At this visit, which is held at 11 + 0 to 13 + 6 weeks’ gestation, we record maternal characteristics and medical history and perform an ultrasound scan for: first, determination of gestational age from the measurement of CRL [19]; second, diagnosis of major fetal abnormalities [20]; and third, measurement of fetal nuchal translucency thickness for assessment of risk for trisomies [21]. In the twin pregnancies, gestational age was determined from the CRL of the larger twin, and chorionicity was determined from the number of placentas and the presence or absence of the lambda sign at the inter-twin membrane-placental junction [22]. Each twin pregnancy was matched with 10 singleton pregnancies that were examined within 48 hours of the twin pregnancy. Written informed consent was obtained from the women agreeing to participate in the study, which was approved by the NHS Research Ethics Committee.

Details of maternal characteristics and the findings of the assessment at 11–13 weeks were recorded in our database. Patients were asked to complete a questionnaire on maternal age, racial origin (White, Black, South Asian, East Asian, or mixed), method of conception (natural or assisted conception by in-vitro fertilization or use of ovulation drugs), medical history (including pre-pregnancy diabetes mellitus type 1 or 2), family history of diabetes mellitus (first-, second- or third-degree relative with diabetes mellitus type 1 or 2), and obstetric history (parous or nulliparous with no previous pregnancies at or beyond 24 weeks; if parous, we recorded whether any of the previous pregnancies were complicated by GDM). The questionnaire was then reviewed by a doctor together with the patient. The maternal weight and height were measured, and the body mass index was calculated in kg/m^2^.

Data on pregnancy outcome were obtained from the maternity computerized records or the general medical practitioners of the women and were also recorded in our database.

### 2.2. Inclusion and Exclusion Criteria

The inclusion criterion for this study was singleton or twin pregnancy delivering a phenotypically normal neonate at or after 28 weeks’ gestation. We excluded pregnancies with pre-pregnancy diabetes mellitus type 1 or 2 and those ending in termination, miscarriage, or delivery before 28 weeks because they may not have had screening and diagnosis of GDM. We also excluded pregnancies with chromosomal abnormalities or major defects diagnosed prenatally or postnatally, those with twin reversed arterial perfusion (TRAP) sequence, or conjoined twins.

### 2.3. Outcome Measure

The outcome measure was GDM. The diagnosis of GDM was based on a 75-g oral glucose tolerance test (OGTT); the diagnostic criteria were fasting plasma glucose level ≥5.6 mmol/L and/or 2-h plasma glucose level ≥7.8 mmol/L [23]. The screening policies leading to an OGTT changed over the years; from the beginning of the study until October 2018, all women had measurement of random plasma glucose at 24–28 weeks, and OGTT was carried out if the concentration was >6.7 mmol/L. From November 2018 onwards, there is a two-stage screening policy for GDM. First, women with any one risk factor (body mass index >30 kg/m^2^, previous birth of a macrosomic baby weighing >4.5 kg, previous GDM, first-degree relative with diabetes or persistent glucosuria) are offered measurement of glycosylated hemoglobin (HbA1c) at booking and, if the value is ≥5.7%, then they have an OGTT, usually at around 12 weeks’ gestation. Second, in all women at 26–28 weeks’ gestation, plasma glucose level is measured 1–2 h after eating ≥50 g of carbohydrate, and if the concentration is ≥6.7 mmol/L, then OGTT is carried out. During the whole study period, an OGTT is also performed if there is polyhydramnios or the fetus becomes macrosomic. 

### 2.4. Statistical Analysis 

Data were expressed as median (interquartile range (IQR)) for continuous variables and *n* (%) for categorical variables. Students *t*-test and χ^2^-square test or Fisher’s exact test were used for comparing outcome groups for continuous and categorical data, respectively. 

Using data on 75,161 singleton pregnancies, we developed an algorithm for the prediction of GDM in singleton pregnancies based on maternal characteristics and medical history [9]. Here, we use the linear predictor from this model as an offset for a further logistic regression model with terms for dichorionic and monochorionic twin pregnancies. This approach allowed us to both develop an extension for twins to our existing algorithm and to recalibrate the algorithm for singletons. An advantage to this approach over re-fitting the history model is that it allowed us to make good use of information gained from a much larger data set.

The statistical software package R was used for data analyses [24].

## 3. Results

### Study Population 

The inclusion criteria were fulfilled by 13,760 singleton and 1376 twin pregnancies; in the latter, 1111 were dichorionic, and 265 were monochorionic. GDM developed in 517 (3.8%) of the singleton pregnancies and in 66 (6.9%) of the dichorionic and 24 (9.1%) of the monochorionic twin pregnancies. 

The maternal and pregnancy characteristics of the GDM and non-GDM groups in singleton and twin pregnancies are shown in Table 1. In the GDM groups in both singleton and twin pregnancies, the women tended to be older, heavier, and shorter, and there was a higher proportion of Black, South Asian, and East Asian racial origin; conceptions with in-vitro fertilization; history of first- or second-degree relative with diabetes; and previous pregnancies complicated by GDM or higher birth weight of last neonates. 

The logistic regression model for the prediction of GDM from maternal characteristics and elements of medical history is shown in Table 2 and Figure 1. Confidence intervals are provided only for the estimates for twin pregnancies, as other coefficients are treated as fixed constants [9].

The performance of screening for GDM of the new model is summarized in Table 3. In singleton pregnancies, at 10%, 20%, and 40% false-positive rates, the detection rates were 42.8%, 58.0%, and 74.7%, respectively. In twin pregnancies, at the same risk cut-offs corresponding to 10%, 20%, and 40% false-positive rates in singletons, the respective false positive rates were 26.9%, 47.0%, and 71.1%, and the detection rates were 63.3%, 81.1%, and 91.1%.

## 4. Discussion

### 4.1. Main Findings

The study has demonstrated that the risk for GDM in twin pregnancies is higher than in singleton pregnancies after adjustment for elements of demographic characteristics and medical history that are known to be associated with GDM. The study also extended our previously described prediction model for GDM derived from the study of 75,161 singleton pregnancies [9] to also include dichorionic and monochorionic twins. In dichorionic and monochorionic twin pregnancies, the respective odds of GDM are 1.4 times and 2.8 times higher than in singleton pregnancies. In twin pregnancies, at the same risk cut-offs used for screening in singleton pregnancies, both the false-positive and detection rates are considerably higher.

In both singleton and twin pregnancies, the risk for GDM increases with increasing maternal age and weight and birth weight z-score of a baby in a previous pregnancy and is higher in women with a previous pregnancy complicated by GDM; in those with first- or second-degree relative with diabetes mellitus; in women of Black, East Asian and South Asian racial origin; and in pregnancies conceived through the use of ovulation induction drugs and is lower in parous women that did not develop GDM in a previous pregnancy. 

### 4.2. Interpretation of Findings and Comparison with Results of Previous Studies

Our finding that the incidence of GDM is higher in twin than singleton pregnancies is consistent with findings of four previous studies. Schwartz et al. [13] reported that the rate of GDM was 7.7% in 429 twin pregnancies and 4.1% in 29,644 singleton pregnancies. Rauh-Hain et al. [14] reported that the rate of GDM was 3.98% in 533 twin pregnancies vs. 2.32 in 23,056 singleton pregnancies (odds ratio 2.2 after adjustment for age, body mass index, race, weight gain, blood pressure, parity, and smoking status). Weissman et al. [15] reported that the rate of GDM was 10.1% in 515 twin pregnancies vs. 2.9% in 12,382 singleton pregnancies. Hiersch et al. [16] analyzed the data of a large cohort of 266,942 singleton and 3901 twin pregancies and found a significantly higher risk for GDM in twins (relative risk of 1.13, after adjustment for age, parity, ethnicity, body mass index, and method of conception). In contrast, two studies reported that the incidence of GDM in twin pregnancies was not significantly different from that in singleton pregnancies. Buhling et al. [17] reported that the rate of GDM was 3.4% in 89 twin pregnancies and 3.4 % in 178 singleton pregnancies (matched for age, body mass index, parity, and ethnicity). Similarly, Morikawa et al. [18] reported that the rate of GDM was 9.3 % in 86 twin pregnancies and 8.4 % in 344 singleton pregnancies (matched for age and body mass index).

We found that the risk of GDM is higher in monochorionic than in dichorionic twin pregnancies. This finding is consistent with the results of a previous study in twin pregnancies [25]. A possible explanation for this finding is that in monochorionic twin pregnancies, serum plasma protein-A (PAPP-A) is lower than in dichorionic twins [26]. We previously reported that there is an association between low serum PAPP-A at 11–13 weeks’ gestation and both type 2 diabetes mellitus and GDM [27,28,29]. PAPPA is a metalloprotease secreted by the human placenta that modulates insulin-like growth factor (IGF) bioavailability through proteolysis of IGF-binding proteins (IGFBPs) 2, 4, and 5. There is evidence that low concentrations of PAPPA result in impaired proteolysis of adipose tissue IGFBPs, which are up-regulated in adipose depots during pregnancy [30]. In turn, IGF signaling decreases, impairing pregnancy-induced increases in adipocyte number and size as well as tissue vascularization. Inadequate adipose tissue adaptation leads to enhanced insulin resistance and impaired glucose tolerance.

The new model predicts 42.8%, 58.0%, and 74.7% of GDM in singleton pregnancies at 10%, 20%, and 40% false-positive rates, respectively. These detection rates are lower than the respective values of 55%, 68%, and 84% achieved in our previous publication [9]. The most likely explanation for such a drop is that the inclusion criteria for screening for GDM has recently become less stringent so that a higher proportion of the population is now found to be screen positive; consequently, the false-positive rate is higher. 

### 4.3. Strengths and Limitations

The study has major strengths, which include a large sample size, a systematic approach of obtaining obstetric and medical history, and measurement of maternal weight and height. Moreover, we modified a previously reported multivariate logistic model derived from the study of 75,161 singleton pregnancies to include prediction of GDM in twin pregnancies, and we matched each twin pregnancy with 10 singleton pregnancies examined within two days from the twins to minimize the potential risk of bias arising from possible changes of screening policies over the long study period of 10 years. A limitation of the study narrates to the method of detecting pregnancies affected by GDM. Universal testing with a diagnostic OGTT was not performed in all pregnancies, as recommended by the International Association of Diabetes and Pregnancy Study Groups [7], but only in a subgroup with either risk factors for GDM [8] or abnormal results of a random blood glucose level at 24–28 weeks’ gestation. Consequently, it is likely that our non-GDM group contained women who had undiagnosed GDM, and therefore, the screening performance of our method was overestimated. 

## 5. Conclusions

In this study, we extended our logistic regression model for first-trimester prediction of GDM from maternal characteristics and medical history to include terms for dichorionic and monochorionic twin pregnancies. The risk of GDM was higher in monochorionic than in dichorionic twin pregnancies, and this could be the consequence of inadequate adipose tissue adaptation, enhanced insulin resistance, and impaired glucose tolerance due to lower serum PAPP-A.

## Figures and Tables

**Figure 1 jcm-10-03814-f001:**
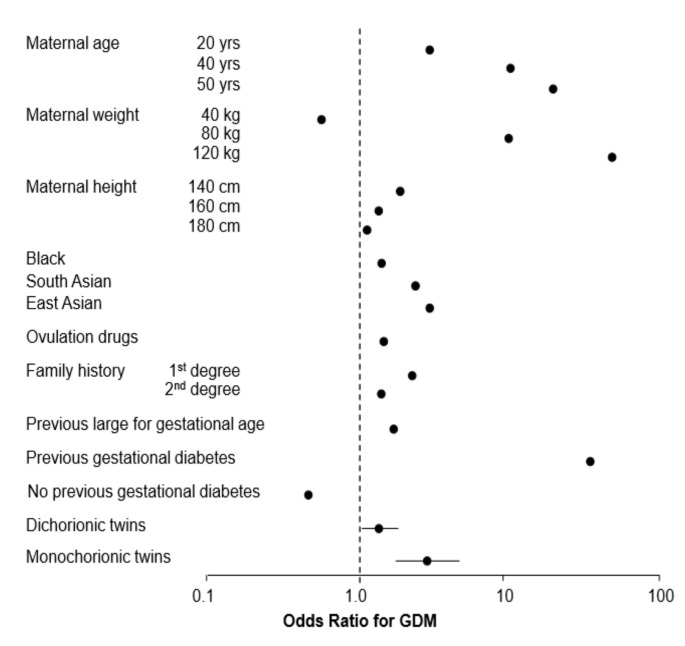
Odds ratio for gestational diabetes mellitus according to maternal factors and chorionicity. The odds ratio for each type of twin pregnancy refers to a woman who is 35 years old, 69 kg in weight, 164 cm in height, is of White racial origin, conceived naturally, has no family history of diabetes, and is nulliparous.

**Table 1 jcm-10-03814-t001:** Demographic and pregnancy characteristics of the study population.

Characteristic	Singleton Pregnancies			Twin Pregnancies		
	No GDM (*n* = 13,243)	GDM (*n* = 517)	*p*-Value	No GDM (*n* = 1286)	GDM (*n* = 90)	*p*-Value
Maternal age (years)	32.5 (28.7, 35.7)	34.0 (30.5, 37.1)	<0.0001	34.25 (30.4, 37.4)	35.5 (31.4, 38.8)	0.068
Maternal weight (kg)	66.5 (59.4, 76.3)	76.1 (65.0, 91.0)	<0.0001	68.8 (61.0, 79.0)	75.0 (64.5, 90.9)	0.0005
Maternal height (cm)	165 (161, 170)	164 (159, 168)	<0.0001	166 (162, 170)	164 (158, 169)	0.003
Body mass index (kg/m^2^)	24.2 (21.8, 27.8)	28.6 (24.3, 33.3)	<0.0001	24.6 (22.1, 28.2)	28.0 (24.6, 33.3)	<0.0001
Gestational age (weeks)	12.7 (12.3, 13.1)	12.7 (12.4, 13.1)	0.015	12.7 (12.3, 13.1)	12.9 (12.4, 13.1)	0.329
Racial origin			<0.0001			0.002
White	9231 (69.7%)	276 (53.4%)		892 (69.4%)	56 (62.2%)	
Black	2735 (20.7%)	132 (25.5%)		285 (22.2%)	21 (23.3%)	
South Asian	550 (4.2%)	58 (11.2%)		40 (3.1%)	10 (11.1%)	
East Asian	252 (1.9%)	26 (5.0%)		21 (1.6%)	0	
Mixed	475 (3.6%)	25 (4.8%)		48 (3.7%)	3 (3.3%)	
Family history of diabetes			<0.0001			0.018
1st degree	1525 (11.5%)	121 (23.4%)		167 (13.0%)	19 (21.1%)	
2nd degree	1231 (9.3%)	56 (10.8%)		141 (11%)	16 (17.8%)	
3rd degree	259 (2.0%)	11 (2.1%)		19 (1.5%)	1 (1.1%)	
Method of conception			0.387			0.107
Natural	12,611 (95.2%)	486 (94.0%)		886 (68.90%)	53 (58.9%)	
In-vitro fertilization	530 (4.0%)	25 (4.8%)		379 (29.5%)	36 (40.0%)	
Ovulation drugs	102 (0.8%)	6 (1.2%)		21 (1.6%)	1 (1.1%)	
Parity			<0.0001			<0.0001
Nulliparous	6532 (49.3%)	210 (40.6%)		678 (52.7%)	46 (51.1%)	
Parous, previous GDM	105 (0.8%)	82 (15.9%)		9 (0.7%)	8 (8.9%)	
Parous, no previous GDM	6606 (49.9%)	225 (43.5%)		599 (46.6%)	36 (40.0%)	
Birthweight of last neonate (g)	3350 (3015, 3689)	3433 (3036, 3860)	0.024	3349 (3000, 3660)	3405 (3178, 3750)	0.195

**Table 2 jcm-10-03814-t002:** Logistic regression model for the prediction of gestational diabetes mellitus from maternal characteristics and elements of medical history. Confidence limits are provided only for the twin pregnancies because the odds ratio for the other terms are identical to those of our previous publication in singleton pregnancies [9].

Term	Odds Ratio (95% CI)	Coefficient	*p*-Value
Intercept	-	−3.53042	<0.0001
Twins: dischorionic	1.3601 (1.0213–1.8115)	0.30759	0.035
Twins: monochorionic	2.7771 (1.7205–4.4823)	1.02139	<0.0001
Previous GDM	50.44	3.92090	
Weight in kg—69	1.0208	0.02060	
Nulliparous or parous with no previous GDM			
Parous: no previous GDM	0.4545	−0.78850	
Age in years—35	1.0841	0.08070	
Weight in kg—69	1.0389	0.03810	
Height in cm—164	0.9426	−0.05910	
1st-degree relative with DM	2.5427	0.93320	
2nd-degree relative with DM	1.7984	0.58690	
Ovulation drugs	1.6019	0.47120	
Black racial origin	1.5780	0.45620	
East Asian racial origin	2.9232	1.07270	
South Asian racial origin	2.3165	0.84010	
Birth weight z-score of previous pregnancy	1.2520	0.22470	

**Table 3 jcm-10-03814-t003:** Performance of screening for gestational diabetes mellitus in singleton and twin pregnancies.

Pregnancy	FPR (%)	DR (LCL–UCL) %
Singleton	10.0	221/517; 42.8 (38.4–47.1)
Twins	26.9	57/90; 63.3 (52.5–73.3)
Singleton	20.0	300/517; 58.0 (53.6–62.3)
Twins	47.0	73/90; 81.1 (71.5–88.6)
Singleton	40.0	386/517; 74.7 (70.7–78.4)
Twins	71.1	82/90; 91.1 (83.2–96.1)

FPR, false-positive rate; DR, detection rate; LCL, lower confidence limit; UCL, upper confidence limit.

## Data Availability

Research data are not shared.

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
