# Peer review of "First-Trimester Screening for Gestational Diabetes Mellitus in Twin Pregnancies"

_jcm, 2021, doi:10.3390/jcm10173814_

Round 1
Reviewer 1 Report
The manuscript submitted by Olga Buerger et. al aimed to investigate several purposes: 1. to compare the rate of GDM in BCBA and MCBA twin pregnancies with singleton pregnancies. Comparison was performed in a 1:10 ratio (twins vs singleton, respectively) 2. To launch on twin pregnancies a prediction model for GDM (similar to a prediction model for GDM in singleton pregnancy). The study population was derived from a prospective screening study on early prediction of pregnancy complications, in women attending for their routine first visit at 11 to 13.6 weeks’ gestation, between 2010 -2020. Findings of the assessment at 11-13 weeks and detailed questionnaire on maternal and pregnancy characteristics as well as pregnancy outcome were recorded. The outcome measure was GDM, diagnosed with a 75-gram OGTT. Notably during the study period, the screening method of identification of GDM has been changed, therefore starting OCT 2018 not all women had the OGTT but only those with risk factors. Included in the study 13,760 singleton and 1,376 twin pregnancies (1,111 BCBA and 265 MCBA). Authors demonstrated that the risk for GDM is higher in twin as compared to singleton pregnancies, while the odds ratio for MCBA twins are higher than for BCBA twins. By logistic regression analysis they demonstrate that in both groups (singleton and twin pregnancies) there were similar predictors of GDM. Additionally in twin pregnancies, using risk cut-offs corresponding to 10% and 20% FP rates in singletons, the respective FP rates in twins were 27% and 47% with DR of 63% and 81% for GDM. Authors concluded that they could extend their logistic regression model for first-trimester prediction of GDM to twin pregnancies.
This is an interesting, well written manuscript that enables us, to use a first trimester prediction model for GDM, in twins' pregnancies "only" from maternal characteristics and medical history. The limitations of the study are explicitly acknowledged, the conclusions are precise and strongly supported by the results. This is a scientifically sound and clinically important paper which merits publication in Journal of Clinical Medicine. I have only few comments to be addressed:
- Could you add data concerning the rate of treatment with Aspirin in the study cohort?
- Could you add a graphic presentation of the prediction model to develop GDM base on several elements of medical history or maternal characteristics in BCBA and MCBA twins? I think it would be very informative for the readers
Author Response
Thank you very much for your comments.
I have only few comments to be addressed:
- Could you add data concerning the rate of treatment with Aspirin in the study cohort?
Response: Thank you. Unfortunately we don’t have information on Aspirin uptake on this cohort.
- Could you add a graphic presentation of the prediction model to develop GDM base on several elements of medical history or maternal characteristics in BCBA and MCBA twins? I think it would be very informative for the readers
Response: Thank you. We have now added Figure 1.
Reviewer 2 Report
Reviewer Comments: JCM-1335099
This is an interesting attempt to perform perinatal gestational diabetes risk assessment for dichorionic and monochorionic twin gestations. However, it adds little to the obstetric literature.
The authors’ data confirm increased maternal body mass index (BMI) is a risk for development of gestational diabetes (GDM). However, the other identified GDM risks, advanced maternal age, ethic variability and family history of type 2 diabetes, for which the study was insufficiently powered to evaluate, often co-exist with increased BMI and therefore serve as uncontrolled variables.
Although an early hemoglobin A1c was only obtained on individuals who were predetermined to be at risk for developing GDM, evaluating the predictive value of glycosylated hemoglobin for GDM might provide useful clinical data from this study.
Author Response
Response: Thank you. As explained in Methods patients were recruited between January 2010 and August 2020. First trimester measurement of HbA1C in a small subgroup of high risk patients was initiated in November 2018. Consequently we do not have HbA1C results for the vast majority of GDM patients.
Round 2
Reviewer 2 Report
Conclusions are reasonable while adding little new information for clinicians managing twin gestations. The inability to obtain hemoglobin A1c levels on all subjects is unfortunate but does not warrant rejection of the manuscript.
Author Response
Response: Thank you.
This manuscript is a resubmission of an earlier submission. The following is a list of the peer review reports and author responses from that submission.